(GIGA)bYte

DATA RELEASE

# Species composition and distribution of the *Anopheles gambiae* complex circulating in Kinshasa

Josue Zanga[1,2,*], Emery Metelo[2,3], Nono Mvuama[2], Victoire Nsabatien[2,4,*], Vanessa Mvudi[2], Degani Banzulu[5], Osée Mansiangi[2], Maxwel Bamba[2], Narcisse Basosila[6], Rodrigue Agossa[7] and Roger Wumba[1]

1 University of Kinshasa, Tropical Medicine Department, Kinshasa, Democratic Republic of the Congo
2 Kinshasa School Public Health, Laboratory of Bio-ecology and Vector Control, Department of Health-Environment, Kinshasa, Democratic Republic of the Congo
3 University of Bandundu, Faculty of Medicine, Bandundu Ville, Democratic Republic of the Congo
4 Laboratory of Bioecology and Vector Control, Department of Environmental Health, Kinshasa School of Public Health, Kinshasa, Democratic Republic of the Congo
5 University of Kinshasa, Department of Neurology, Kinshasa, Democratic Republic of the Congo
6 National Malaria Control Programme, Vector Control Service, Kinshasa, Democratic Republic of Congo
7 Cotonou Entomological Research Center (CREC), Cotonou, Benin

## ABSTRACT

Understanding the distribution of Anopheles species is essential for planning and implementing malaria control programmes. This study assessed the composition and distribution of cryptic species of the main malaria vector, the *Anopheles gambiae* complex, in different districts of Kinshasa. Anopheles were sampled using CDC light traps in the four Kinshasa districts between July 2021 and June 2022, and then morphologically identified. Equal proportions of *Anopheles gambiae* s.l. per site were subjected to polymerase chain reaction to identify the cryptic species of the *Anopheles gambiae* complex. *Anopheles gambiae* complex specimens were identified throughout Kinshasa. The average density significantly differed inside and outside households. Two species of this complex circulate in Kinshasa: *Anopheles gambiae* and *Anopheles coluzzii*. In all the study sites, *Anopheles gambiae* was the most widespread species. Our results provide an important basis for future studies on the ecology and dynamics of cryptic species of the *Anopheles gambiae* complex in Kinshasa.

**Subjects** Ecology, Biodiversity, Taxonomy

**Submitted:** 25 October 2023

\* Corresponding authors. E-mail: josuezanga1979@gmail.com; vnsabatien@gmail.com

Preprint submitted at https://doi.org/10.1101/2023.10.26.564181

Included in the series: *Vectors of human disease* (https://doi.org/10.46471/GIGABYTE_SERIES_0002)

# DATA DESCRIPTION

## Background and context

The genus *Anopheles* is by far the Culicidae vector most targeted by vector control efforts. This vector is responsible for transmitting malaria, an infectious disease caused by protozoan parasites of the genus *Plasmodium*, transmitted by female *Anopheles* mosquitoes [1]. Of the 465 officially recognised species of *Anopheles* mosquitoes, about 70 can transmit malaria parasites to humans [2]. In Africa, the main malaria vectors fall within four taxonomic categories: the *Anopheles gambiae* complex, *Anopheles funestus* gpe, *Anopheles nili* gpe and *Anopheles moucheti* gpe [3, 4]. These species complexes or groups have different distributions, behaviours and ecology [3, 4].

Malaria transmission in sub-Saharan Africa is largely carried out by the *Anopheles gambiae* complex [2]. This complex comprises about nine related and morphologically indistinguishable subspecies: *Anopheles gambiae*, *Anopheles coluzzii*, *Anopheles arabiensis*, *Anopheles melas*, *Anopheles merus*, *Anopheles bwambae*, *Anopheles quadriannulatus*, *Anopheles amharicus* and *Anopheles fontaneilli* [5].

The distribution and density of these species are clearly influenced by the environmental conditions and climatic characteristics of the region [6]. *Anopheles arabiensis* is predominant in the driest areas. Despite being anthropophilic, it strongly tends to be zoophilic. However, due to the ever-changing human environment and the expansion of African cities, *Anopheles arabiensis* is increasingly present in large cities [7, 8]. *Anopheles melas* and *Anopheles merus* are also important vectors [1, 9]. *Anopheles melas* is restricted to coastal areas of West and Central Africa, where its larvae develop in brackish water [10]. However, this species is not anthropophilic and has a short life span, making it a poor vector of malaria [11]. *Anopheles merus* is isolated in the coastal region of East Africa [1]. *Anopheles gambiae* and *Anopheles coluzzii* are sympatric across much, but not all, of their African ranges [12]. *Anopheles coluzzii* is usually associated with permanent breeding sites, often created by human activities such as irrigation, rice cultivation and urbanisation. On the other hand, *Anopheles gambiae* is associated with temporary rain pools and puddles [13, 14]. There is no strong evidence that the other species of the *An. gambiae* complex play a role in malaria transmission [15, 16].

In the Democratic Republic of Congo (DRC), the *Anopheles gambiae* complex plays a major role in malaria transmission [17]. Specifically, most publications reported a clear predominance of the former ss form, currently recognised as *Anopheles gambiae* [18–21]. *Anopheles gambiae* is widely distributed in Kinshasa [18, 20]; however, the Bobanga study, conducted in 2013, noted the presence of the sympatric species *Anopheles coluzzii* in the Mont Amba district of Kinshasa [22].

Despite the high ecological diversity and the steady invasion of members of the complex into the city of Kinshasa, no updated studies on the composition and distribution of vectors of the *An. gambiae* Complex across this megacity are available. However, the identification of species within the *An. gambiae* Complex is essential for a proper evaluation of malaria vector ecology and control programmes [23]. Furthermore, understanding the genetic structure of members of the *An. gambiae* Complex is important for addressing important biological and public health issues, such as evolution, the spread of insecticide resistance alleles and the epidemiology of vector-borne diseases [24].

This study aims to establish the composition and distribution of cryptic species of the main malaria vector, *Anopheles gambiae* complex, circulating in different districts of Kinshasa city.

## METHODS

### General spatial coverage

This study was conducted in Kinshasa for 12 months, including the dry and rainy seasons (July 2021–June 2022). Kinshasa is located in Central Africa, on the right bank of the Congo River, between 4°19′30″ South latitude and 15°19′20″ East longitude. The climate is hot and humid (AW4 according to Koppen's classification), with a rainy season from October to May and a dry season from May to September [25].



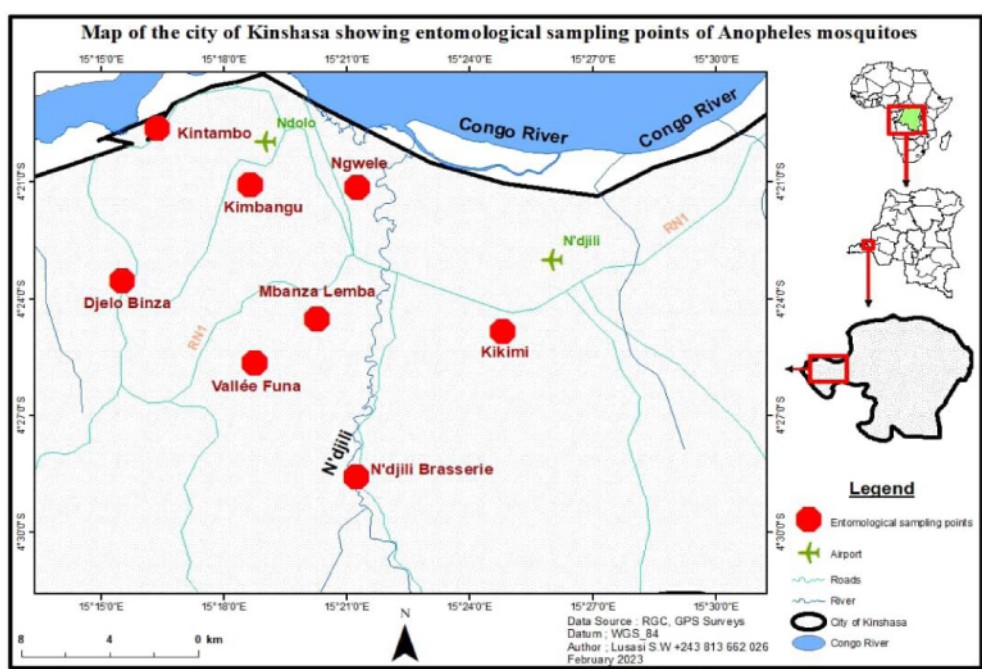

**Figure 1.** Maps of the city of Kinshasa showing the sampling points of *Anopheles* mosquitoes.

Kinshasa is a megalopolis of 9,965 km$^2$. It is administratively subdivided into four districts: Tshangu, Funa, Mont Amba and Lukunga. Each district is subdivided into communes and the communes are subdivided into neighbourhoods. Malaria transmission is perennial, with three strata of transmission [26]. The stratum of low transmission risk is located in the north-central part of Kinshasa (malaria prevalence ≤ 5%); the stratum of intermediate transmission risk is located in the south-central part of Kinshasa (malaria prevalence between 5% and 30%); finally, the stratum of high transmission risk is located in the south-western and eastern parts of the city of Kinshasa (malaria prevalence > 30%).

To study the distribution of members of the *An. gambiae* Complex, the *Anopheles* spp. were sampled in the three malaria transmission strata in the city of Kinshasa [26]. In each stratum, two sites (or neighbourhoods) from two different communes were selected. The selection of these sites (Figure 1) was based on the administrative distribution of this megalopolis, focusing on its four districts:

- Funa District: Kimbangu site (S 04°20′51″S, E 15°19′12′) in the commune of Kalamu and Valée de Funa site (S 04°24′49″S, E 15°16′52″) in the commune of Selembao. These two sites are located in the central-northern part of the city. They exhibit a variable risk of malaria transmission (prevalence ≤ 5% for the commune of Kalamu and between 5% and 30% for the commune of Selembao) [26];
- Mont Amba District: Ngwele site (S 04°20′59″, E 15°20′17″) in the commune of Limete and Mbanza Lemba site (S 4°24′59″; E 15°19′27″) in the commune of Lemba. These sites exhibit an intermediate risk of malaria transmission (prevalence between 5% and 30%) [26];
- Lukunga District: Djelo Binza site (S 04°24′47″, E 15°20′47″) in Ngaliema commune and Kitambo site (S 04°19′37″, E 15°16′22″) in the commune of Kitambo. These two sites are

located in the western part of the city and present an intermediate risk of malaria transmission (prevalence between 5% and 30%) [26];

- Tshangu District: Kikimi site (S 4°24′51″; E 15°24′48″) in the commune of Kimbanseke and Ndjii Brasserie site (S 4°28′35″; E 15°21′14″) in the commune of Nsele. These two sites are located in the eastern part of the town and present a high risk of malaria transmission (prevalence > 30%) [26].

In each study site, mosquito sampling was performed using Center for Disease Control (CDC) light traps.

## CDC Light trap catches (CDC light)

In each site, ten households were selected (with the agreement of the heads of the households) for the collection of adult mosquitoes using CDC light traps. This collection was conducted overnight between 18:00 and 06:00, once a month throughout the entire study period. Two traps were used in each household, one placed inside and the other outside. The outdoor traps were placed within five metres of the front door. For each capture point, temperature and humidity were measured using ThermoPro thermohygrometers (ThermoPro, Lawrenceville, USA).

To optimise the genetic diversity of the Culicidae species collected in each study site, households selected to capture Culicidae by CDC light traps were sampled along a 15 km long transect. The distribution of ten selected households per site followed the following pattern: three households between 0–3 m, three between 6–9 km, and four between 12–15 km. Different houses were used for each night, with the consent of the heads of the households. Captured Culicidae were sent to the Bioecology and Vector Control Laboratory of the Kinshasa School of Public Health for morphological identification using the Coetzee key [27].

## Molecular identification of collected samples

After morphological identification, an equal proportion of *Anopheles gambiae* complex, sampled per site, according to transect extent, were individually preserved in 1.5 mL Eppendorf tubes with silica gels. The samples were then sent to the Entomological Research Center in Cotonou, Benin, to identify cryptic species within the *An. gambiae* Complex using polymerase chain reaction (PCR).

After extraction of the genomic DNA, the amplification and determination of the species of the *An. gambiae* complex were carried out according to the protocol outlined by Scott *et al.* [28]. For this purpose, the following primers were used: UN: GTGTGCCGCTTCCTCGATGT; AG: CTGGTTTGGTCGGCACGTTT; AA: AAGTGTCCTTCTCCATCCTA; ME: TGACCAACCCACTCCCTTGA; QD: CAGACCAAGATGGTTAGTAT.

The UN primer binds to the same position in the rDNA of all five species, while AG binds specifically to *Anopheles gambiae*. ME binds to both *Anopheles merus* and *Anopheles melas*. Finally, AA binds to *Anopheles arabiensis* and QD to *Anopheles quadriannulatus*.

The Santolamazza protocol allows for a better differentiation of the molecular forms of *Anopheles gambiae,* relying on the specific and irreversible insertion of a 230 bp transposon (SINE200). This transposon is present on the X chromosome of *Anopheles coluzzii*, whereas it is absent in *Anopheles gambiae.* This genetically inherited feature allows for the unambiguous, simple and direct recognition of *Anopheles gambiae* and

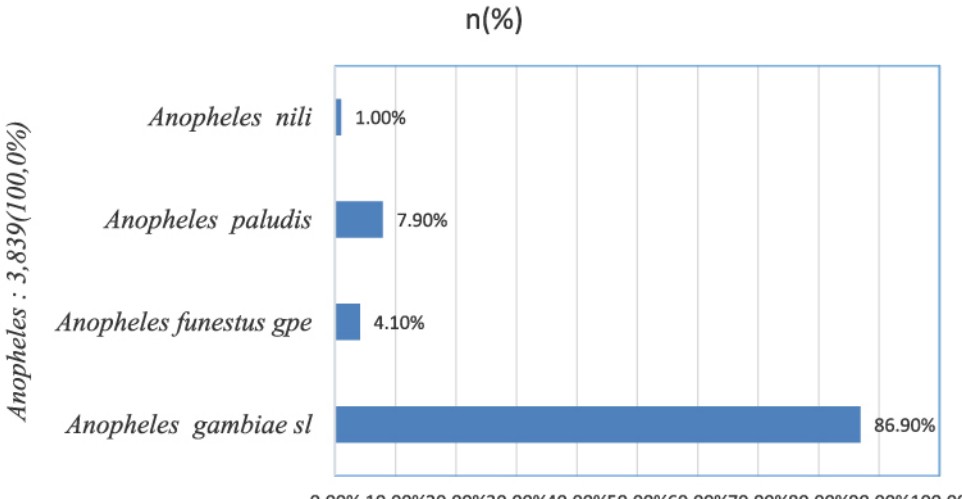

**Figure 2.** Distribution of *Anopheles* species from Kinshasa captured by the CDC light technique during our study.

*Anopheles coluzzii* [29]. In addition to the classical PCR components, the following primers were used: 200X6.1F-TCGCCTTAGACCTTGCGTTA and 200X6.1R-CGCTTCAAGAATTCGAGATAC.

## Data analysis

Data were analysed using Origin version 6.1-Scientific graphing and Data Analysis Software [30]. Relationships between mean densities were analysed using the *t*-student test. The test of significance was conducted with a significance level set at 0.05. A *p*-value less than 0.05 was considered significant during the analysis. Molecular diagnosis of the species was based on the primer set S200 X6.1 in *Anopheles gambiae/coluzzii,* hybrid form; Ac, *Anopheles coluzzii* (479 bp); Ag, *Anopheles gambiae* (249 bp); nc, negative control; L, scale = 100 bp (Solis BioDyne).

## RESULTS

## Composition of *Anopheles* spp. fauna in Kinshasa and nature of identified larval sites

During the whole period of our study, 3,839 *Anopheles* were collected using CDC light traps. After morphological identification, four species of the genus Anopheles were identified: *Anopheles gambiae* complex (11.04%), *Anopheles funestus* gpe (0.53%), *Anopheles paludis* (1.01%) and *Anopheles nili* complex (0.13%) (Figure 2).

While *Anopheles gambiae* complex was identified in all sites, *Anopheles nili* and *Anopheles paludis* were limited to the Tshangu district (Table 1). Apart from *Anopheles paludis*, which was caught more often outside houses, the other *Anopheles* species were predominantly caught inside the sampled houses. Compared to the other identified *Anopheles* species, a significant difference in mean density was observed between the *Anopheles gambiae* complex captured by CDC light trap inside (160.33 ± 40.41) and outside (116.50 ± 20.17) households in Kinshasa ($t$ = 3.36; $p$ = 0.002).

**Table 1.** Distribution of the Anopheline species captured by the CDC light technique in the four districts of Kinshasa from July 2021 to June 2022.

| Districts | Sites | Species | CDC light traps | | | | | |
|---|---|---|---|---|---|---|---|---|
| | | | Indoor | *n* (%) | Outdoor | *n* (%) | *n* | *n* (%) |
| MONT AMBA | Ngwele | *An. gambiae* complex | 283 | 7.37 | 216 | 5.63 | 499 | 13.00 |
| | | *An. funestus* gpe | 8 | 0.21 | 7 | 0.36 | 15 | 0.39 |
| | Mbanza Lemba | *An. gambiae* complex | 186 | 4.85 | 115 | 5.93 | 301 | 7.84 |
| | | *An. funestus* gpe | 12 | 0.31 | 9 | 0.46 | 21 | 0.55 |
| TSHANGU | Kikimi | *An. gambiae* complex | 312 | 8.13 | 245 | 12.64 | 557 | 14.51 |
| | | *An. funestus* gpe | 15 | 0.39 | 7 | 0.36 | 22 | 0.57 |
| | | *An. paludis* | 103 | 2.68 | 137 | 7.07 | 240 | 6.25 |
| | | *An. nili* | 9 | 0.23 | 9 | 0.46 | 18 | 0.47 |
| | Ndjili Brasserie | *An. gambiae* complex | 292 | 7.61 | 239 | 12.33 | 531 | 13.83 |
| | | *An. funestus* gpe | 37 | 0.96 | 27 | 1.39 | 64 | 1.67 |
| | | *An. paludis* | 29 | 0.76 | 36 | 1.86 | 65 | 1.69 |
| | | *An nili* | 11 | 0.29 | 8 | 0.41 | 19 | 0.49 |
| FUNA | Kimbangu | *An. gambiae* complex | 89 | 2.32 | 40 | 1.04 | 129 | 3.36 |
| | | *An. funestus* gpe | 6 | 0.16 | 3 | 0.08 | 9 | 0.23 |
| | Valée Funa | *An. gambiae* complex | 208 | 5.42 | 154 | 4.01 | 362 | 9.43 |
| | | *An. funestus* gpe | 6 | 0.16 | 5 | 0.13 | 11 | 0.29 |
| LUKUNGA | Djelo Binza | *An. gambiae* complex | 286 | 7.45 | 199 | 5.18 | 485 | 12.63 |
| | | *An. funestus* gpe | 8 | 0.21 | 5 | 0.13 | 13 | 0.34 |
| | Kitambo | *An. gambiae* complex | 278 | 7.24 | 196 | 5.11 | 474 | 12.35 |
| | | *An. funestus* gpe | 2 | 0.05 | 2 | 0.05 | 4 | 0.10 |
| TOTAL | | | 2,180 | 56.7856 | 1,659 | 64.6295 | 3,839 | 100 |

## Monthly dynamics of *Anopheles gambiae* complex sampled in Kinshasa

The density dynamics of *Anopheles* spp. were variable monthly during the study period. The density of anopheline mosquitoes captured by CDC light traps was very high in November (10.73%) and April (10.24%), whereas lower densities were noted in July (6.69%) and February (6.20%). The monthly proportion of *Anopheles gambiae* complex caught during our study was high in November (9.35%) and April (9.20%). The peak of other *Anopheles* species, distinct from the *Anopheles gambiae* complex, was noted in January when 1.67% of all *Anopheles* spp. was captured (Figure 3).

## Molecular identification of the collected *Anopheles gambiae* complex

A total of 100 specimens of the *Anopheles gambiae* complex per site, captured by CDC light traps, were selected. The molecular identification results of the different species of the

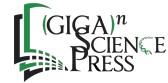

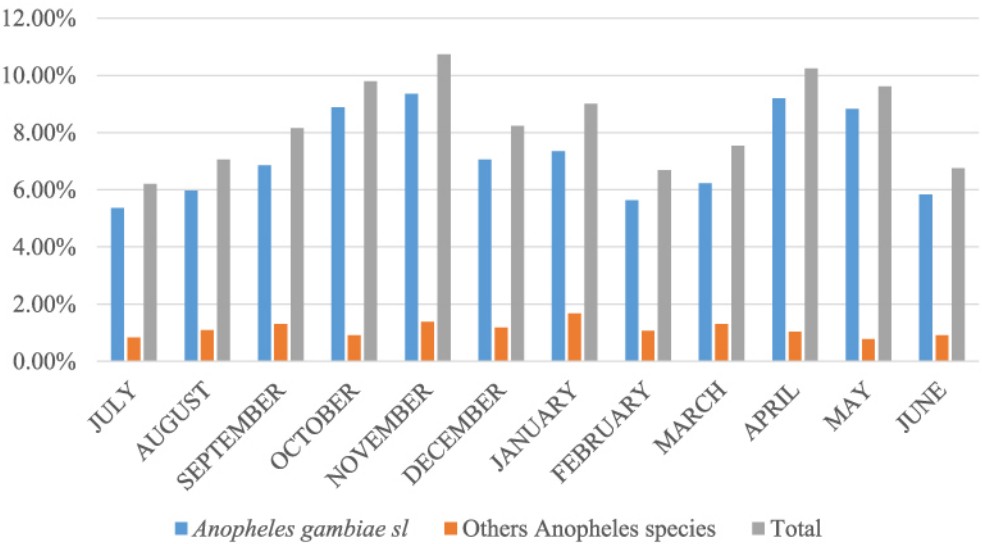

**Figure 3.** Monthly distribution of *Anopheles gambiae* complex captured by CDC light during our study.

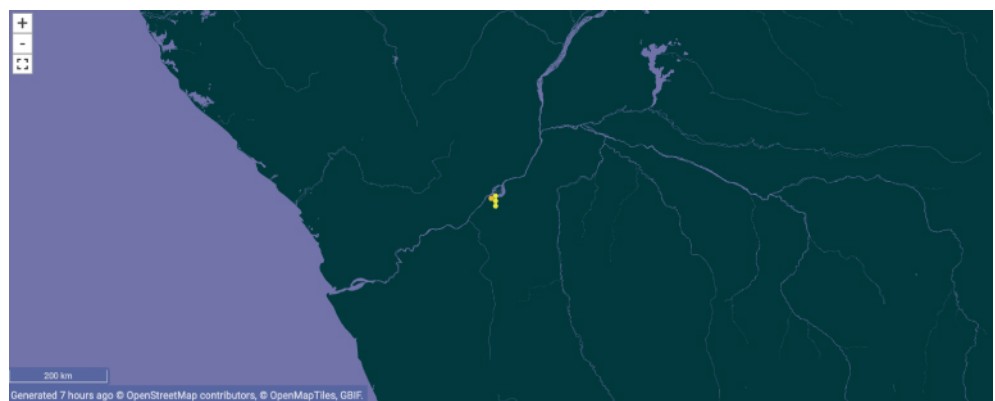

**Figure 4.** Interactive map of the georeferenced occurrences hosted by GBIF [31]. https://www.gbif.org/dataset/8662eca7-981d-4e75-8ed4-d957644b20bc

*Anopheles gambiae* complex are shown in Table 2. Two species of this complex circulate in Kinshasa: *Anopheles gambiae* and *Anopheles coluzzii*. The amplification and determination of members of the *Anopheles gambiae* complex was performed using a combination of the Scott and Santolamazza protocols. The migrations shown in Figure 4 enabled *Anopheles gambiae* (249 bp) and *Anopheles coluzzii* (479 bp) to be distinguished [31]. In all study sites, *Anopheles gambiae* was the main species, with 674 specimens out of the 800 *Anopheles gambiae* complex that passed the molecular identification (Figure 5).

## RE-USE POTENTIAL

The identification of malaria vector species circulating in a given environment is essential for the proper evaluation of control programmes. Four species of the genus *Anopheles* were identified throughout the provincial city of Kinshasa between July 2021 and June 2022. These are *Anopheles gambiae* complex, *Anopheles funestus* gpe, *Anopheles paludis* and



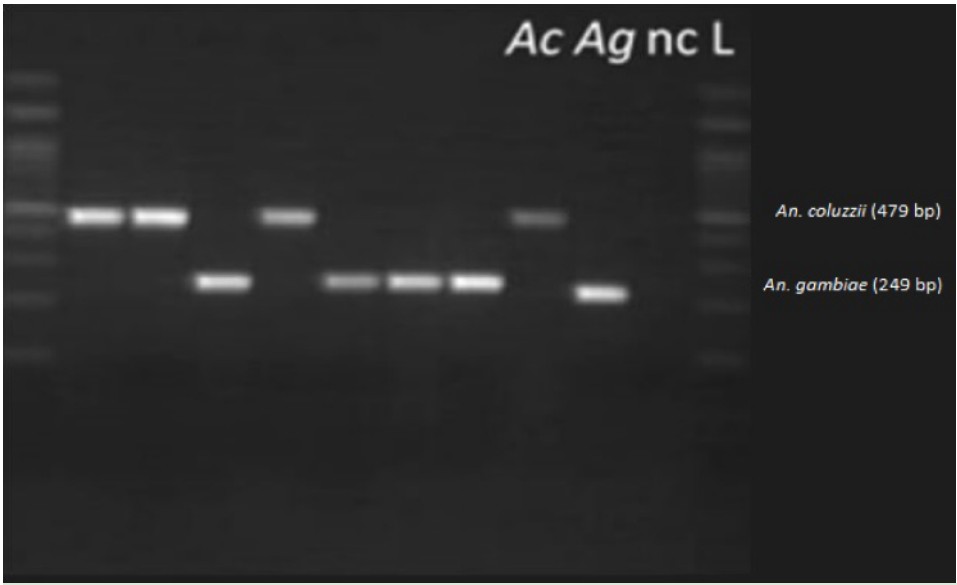

**Figure 5.** Gel image of PCR amplification of the species specific IGS (intergenic spacer) region of members of the *Anopheles gambiae* complex. Legend: Ac, *Anopheles coluzzii*; Ag, *Anopheles gambiae*; nc, negative control; L, scale.

**Table 2.** Molecular identification of cryptic species of *Anopheles gambiae* s.l. circulating in Kinshasa.

| Districts | Sites | N | Cryptic species | | |
|---|---|---|---|---|---|
| | | | *An. gambiae* | *An. coluzzii* | Hybride |
| Mont AMBA | Ngwele | 100 | 68 | 32 | 0 |
| | Mbanza Lemba | 100 | 90 | 10 | 0 |
| TSHANGU | Kikimi | 100 | 63 | 37 | 0 |
| | Ndjili Brasserie | 100 | 79 | 21 | 0 |
| FUNA | Kimbangu | 100 | 100 | 0 | 0 |
| | Valée Funa | 100 | 82 | 18 | 0 |
| LUKUNGA | Djelo Binza | 100 | 92 | 8 | 0 |
| | Kintambo | 100 | 100 | 0 | 0 |
| Total | | 800 | 674 | 126 | 0 |

*Anopheles nili*. Among all Anophelinae, the *Anopheles gambiae* complex was the most important species, followed by *Anopheles paludis* and *Anopheles funestus* gpe. These results are consistent with the conclusions of a previous study by Karch *et al.* [32], who found a high predominance of *Anopheles gambiae* sl among the Anopheles circulating in Kinshasa. Riveron *et al.* in 2018 [18] and Zanga *et al.* in 2022 [20], working in the east of the city of Kinshasa and across seven sites spread over the city of Kinshasa, respectively, revealed a clear predominance of the *Anopheles gambiae* complex. By monitoring the monthly dynamics of the genus *Anopheles*, we showed a fluctuation in their respective frequencies during the study period. Compared to the other species that did not show significant fluctuations, the frequencies of *Anopheles gambiae* complex increased rapidly during the rainy season, with maximum densities recorded in November and April, then decreased

steadily during the dry season, with lower densities in June and July. The fluctuation in rainfall between dry and wet seasons in Kinshasa, coupled with the presence of temporary puddles (shallow, sunny pools) widely distributed in Kinshasa during the rainy season [20], correlated with the high densities of the *Anopheles gambiae* complex [33, 34].

The molecular identification of specimens of *Anopheles gambiae* complex captured by CDC light showed that *Anopheles gambiae* and *Anopheles coluzzii* are the only species of the *An. gambiae* complex circulating in Kinshasa. In Kinshasa, these two species were previously identified by Riveron *et al.* and Zanga *et al.* [18, 20].

*Anopheles gambiae* was found in most of the selected Kinshasa districts and study sites. Apart from the Lukunga district, *An. coluzzii* was present throughout the city of Kinshasa. Its presence was very high in the Tshangu District (Kikimi and Ndjili Brasserie sites) compared to the Mbanza Lemba and Valée de Funa sites in the Mont Amba and Funa District. *Anopheles coluzzii* was also found in high proportions at the Ngwele site. In 2018, Riveron *et al.* also reported the coexistence of these two sympatric species in Kinshasa, specifically in N'Djili Brasserie, within the Tshangu district.

The distribution of the species involved in malaria transmission identified in this study provides an important basis for better defining and directing vector control measures in this megacity. However, understanding the influence of ecological features on the distribution of these species and determining the infectivity of these vectors are still essential. Therefore, more in-depth studies on this issue should be considered.

## DATA VALIDATION AND QUALITY CONTROL

Mosquitoes were identified by experienced taxonomists using a well-established morphological key [27] and molecular techniques [28, 29]. The final dataset was validated in the Integrated Publishing Toolkit (IPT) of the Global Biodiversity Information Facility (GBIF) [35]. The IPT ensures data validation through its network. Metadata fields are available on the dataset page in GBIF [31].

## DATA AVAILABILITY

The data supporting this article are published through the Integrated Publishing Toolkit of GBIF [35] and are available under a CC0 waiver from GBIF [31].

## EDITOR'S NOTE

This paper is part of a series of Data Release articles working with GBIF and supported by TDR, the Special Programme for Research and Training in Tropical Diseases hosted at the World Health Organization [36].

## ABBREVIATIONS

CDC, Center for Disease Control; DRC, Democratic Republic of Congo; GBIF, Global Biodiversity Information Facility; IPT, Integrated Publishing Toolkit; PCR, polymerase chain reaction.

## DECLARATIONS

### Ethics approval and consent to participate

The authors declare that ethical approval was not required for this type of research.

## Competing interests

The authors declare that they have no competing interests.

## Authors' contributions

JZ, EM and RW designed and implemented the study. JZ, OM, VN, MB and NB were responsible for collecting the data. JZ, FA and NM performed the statistical analysis and prepared the manuscript for publication. VN contributed to data curation, visualization and formal analysis on excel. All the authors helped write the manuscript. FA, DB and RW read and edited the manuscript before submission.

## Authors' information

JZ, NM, VN, OM and MB are involved in vector surveillance and control, and ITNs durability at national level, and are all researchers in the Bioecology and Vector Control Laboratory at the Kinshasa School of Public Health (University of Kinshasa). EM and FA carry out vector mapping and resistance monitoring activities at the national level, and are all researchers in the Entomology Unit of the National Institute of Biomedical Research. FA is also the chef of a party for the PMI Evolve project in the DRC and a researcher at the Cotonou Entomological Research Center (CREC), Cotonou, Benin. During this study, EM was also Professor at the University of Bandundu, Faculty of Medicine, Bandundu Ville, Democratic Republic of Congo. NB is conducting studies on the mapping of Anopheles vectors of malaria at the national level and monitoring invasive species, and is the focal point for the National Malaria Control Program. DB and RW are entomology researchers at the University of Kinshasa, Department of Neurology, Kinshasa, Democratic Republic of the Congo.

## Funding

Material and financial support for this study was provided by the Bioecology and Vector Control Unit of the Environmental Health Department of the Kinshasa School of Public Health. No external funding or financial support was obtained.

## Acknowledgements

We would also like to thank the technicians and researchers at the Bioecology and Vector Control Laboratory at the Kinshasa School of Public Health. We would also like to thank the molecular biology laboratory at the CREC/Benin. We would also like to thank Paloma Helena Fernandes Shimabukuro and Tsiky Rabetrano for their help with data submission and facilitation in using the GBIF platform. In addition, many thanks to Dear Willy LUSASI for his willingness to design the Map of the city of Kinshasa showing the sampling points of Anopheles mosquitoes for this study.

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
