## [Reviewer Report]

Comments on revised manuscriptThe comments from the authors address each of my concerns and I am happy to approve this manuscript for publication.

---

## [Editor Report]

Editor’s AssessmentUnderstanding the distribution of Anopheles mosquito species is essential for planning and implementing malaria control programmes, a task undertaken in this study that assesses the composition and distribution of the Anopheles in different districts of Kinshasa in the Democratic Republic of Congo. Mosquitoes were collected using CDC light traps, and then identified by morphological and molecular means. In total 3,839 Anopheles were collected, and data was digitised, validated and shared via the GBIF database under a CC0 waiver. The project monitoring the monthly dynamics of four species of Anopheles, showing a fluctuation in their respective frequencies during the study period. Review improved the metadata by adding more accurate date information, and this data can provide important information for further basic and advanced studies on the ecology and phenology of these vectors in West Africa.

---

## [Reviewer Report]

Reviewer name and names of any other individual's who aided in reviewer Paul TaconetDo you understand and agree to our policy of having open and named reviews, and having your review included with the published papers. (If no, please inform the editor that you cannot review this manuscript.)YesIs the language of sufficient quality?YesPlease add additional comments on language quality to clarify if needed
Are all data available and do they match the descriptions in the paper? NoAdditional Comments1/ The CDC light trap catch data are available in the GBIF release, but the larva collection data are not included in the release. These larva collection data should be either included in the GBIF release, or it should be made clear in the manuscript that this data is not published.  2/ in the dataset, the data are indicated to be reported at the species level (taxonRank = Species) but there are no An. coluzzii reported. However, in table 3 of the manuscript, some An. coluzzii are reported. This is inconsistent. My guess is that the data reported in the dataset are those out of the morphological identification, hence for An. gambiae at the COMPLEX level, and not the species. This shoud in any case be clarified and corrected : are the data in the dataset provided at the complex or at the species level ? If complex, the ScientificName and taxonRank columns should be corrected. In addition, in the dataset, you could add an "identificationRemarks" column providing the source of identification (morphological or molecular).  3/ in the dataset, for the species scientific name, I suggest to use the names as presented in : Harbach, R.E. 2013. Mosquito Taxonomic Inventory, https://mosquito-taxonomic-inventory.myspecies.info/ . Or at least, to provide the "nameAccordingTo" column.   4/ The data available are of type 'occurrence' ( only in 1 file - the "occurrence" file). For a better presentation of the data and in order to be in line with the GBIF data architecture, I would suggest to transform them into "sampling event" data (consisting in 1 'event core' file, 1 'occurence' file, and potentially extension files), which is more suited to this kind of data acquired from sampling events (see https://ipt.gbif.org/manual/en/ipt/latest/sampling-event-data) and containing external measurements (eg. temperature, see next point). This would enable the user to quickly understand the dates and locations of the sampling events.  5/ Temperature and humidity are included in the main 'occurence' file (column "dynamicProperties") :  - to which reality these data correspond (mean during the night of collection ? ), and how were these data collected (instrument, etc.) ? this information is not provided in the manuscript. - Instead of putting this data in the "occurence" file, I would suggest to add a "measurement" file in the GBIF data release, containing these meteorological data. Doing so would enable to include metadata about these measurements (instrument, etc.) See e.g. https://www.gbif.org/sites/default/files/gbif_IPT-sample-data-primer_en.pdf page 6   6/ in the dataset, for some of the collected mosquitoes, you put "organismRemarks" = "unfed" . How did you collect this information ? I could not see any mention to this feeding identification, neither in the manuscript nor in the dataset.  7/ in the dataset, in the column "SamplingProtocol", there are spelling errors -> "CDC ligth trap cathes" should be corrected to "CDC light trap catches "Are the data and metadata consistent with relevant minimum information or reporting standards? See GigaDB checklists for examples <a href="http://gigadb.org/site/guide" target="_blank">http://gigadb.org/site/guide</a>NoAdditional CommentsSee comments aboveIs the data acquisition clear, complete and methodologically sound?NoAdditional CommentsSee comments aboveIs there sufficient detail in the methods and data-processing steps to allow reproduction?YesAdditional CommentsIs there sufficient data validation and statistical analyses of data quality? YesAdditional CommentsIs the validation suitable for this type of data?YesAdditional CommentsIs there sufficient information for others to reuse this dataset or integrate it with other data?YesAdditional CommentsAny Additional Overall Comments to the AuthorThanks for this nice work and the effort you put to open your data. See comments below and above to improve the work.  1/ comments for figure 1 (map) : the background layer is not very appropriate, as we miss landscape context. Maybe better to put an Open Street Map background layer, or a satellite image.RecommendationMajor Revision

---

## [Reviewer Report]

Upload additional filesDRR-202310-03/form/Data-Review-of-DRR-202310-03.pdfReviewer name and names of any other individual's who aided in reviewer Christopher HunterDo you understand and agree to our policy of having open and named reviews, and having your review included with the published papers. (If no, please inform the editor that you cannot review this manuscript.)YesIs the language of sufficient quality?YesPlease add additional comments on language quality to clarify if needed
Are all data available and do they match the descriptions in the paper? NoAdditional CommentsThe larva data are not included in the GBIF dataset. Some of the descriptions of the data in the manuscript do not match the data available from GBIF.Are the data and metadata consistent with relevant minimum information or reporting standards? See GigaDB checklists for examples <a href="http://gigadb.org/site/guide" target="_blank">http://gigadb.org/site/guide</a>YesAdditional CommentsIs the data acquisition clear, complete and methodologically sound?YesAdditional CommentsIs there sufficient detail in the methods and data-processing steps to allow reproduction?YesAdditional CommentsIs there sufficient data validation and statistical analyses of data quality? YesAdditional CommentsIs the validation suitable for this type of data?YesAdditional CommentsIs there sufficient information for others to reuse this dataset or integrate it with other data?YesAdditional CommentsAny Additional Overall Comments to the AuthorMajor comments (Author action required): 1 - The manuscript describes larva collection and molecular identification of those species, but I cannot see any indication that those data are included in the GBIF dataset. Please clarify whether they are included or not, and if not please add them.  2 - The numbers cited in Table 1 do not match those shown in the GBIF dataset, e.g. the total of indoor/outdoor sampling events quoted in MS table 1 = 2180 / 1659, whereas in GBIF dataset there are 2304 indoor and 1535 outdoor sites listed? Please check your calculations and/or the data submitted to GBIF.  Minor comments (Author action suggested): 1 - There are 59 events in the GBIF data that do not have a date. Please check those data and update if you have those dates available.  2 - The events are all included in the GBIF sampling event dataset, however “individualCount” data are not included, please explain why those counts are not included as observation dataset(s)? i.e. why is there no number of individual mosquitos included in the dataset?  3 - The full DwC-GBIF dataset does include an indication of the indoor/outdoor location of the sampling sites in the "eventRemark" column, but if you are making updates to the dataset may I suggest using the column heading “habitat” to include that information in GBIF either instead or as well.  4 - Ideally, the molecular identification data should be shared. I dont have access to the “protocol of Scott [29]” but my assumption is that the PCR products are differentiated by size via running on a gel? If so, and you have the digital images of those gels please let the GigaByte editors know and they will help you share them via the GigaDB database.  Please see the attached file "Data-Review-of-DRR-202310-03.pdf" for more details about the above concerns.
RecommendationMajor Revision